# Cardiac Magnetic Resonance Imaging and Transthoracic Echocardiography: Investigation of Concordance between the Two Methods for Measurement of the Cardiac Chamber

**DOI:** 10.3390/medicina55060260

**Published:** 2019-06-09

**Authors:** Muhammet Gürdoğan, Fethi Emre Ustabaşıoğlu, Osman Kula, Selçuk Korkmaz

**Affiliations:** 1Department of Cardiology, School of Medicine, Trakya University, Edirne 22030, Turkey; 2Department of Radiology, School of Medicine, Trakya University, Edirne 22030, Turkey; ustabasioglu@hotmail.com (F.E.U.); dr.osmankula@gmail.com (O.K.); 3Department of Biostatistics and Medical Informatics, School of Medicine, Trakya University, Edirne 22030, Turkey; selcukorkmaz@gmail.com

**Keywords:** cardiac magnetic resonance, transthoracic echocardiography, chamber, measurement, concordance

## Abstract

*Background and objectives:* Cardiac magnetic resonance (CMR) imaging is the gold standard method for the detection of ventricular volumes and myocardial edema/scar. Transthoracic echocardiography (TTE) imaging is primarily used in the evaluation of cardiac functions and chamber dimensions. This study aims to investigate whether the chamber diameter measurements are concordant with each other in the same patient group who underwent TTE and CMR. *Materials and Methods:* The study included 41 patients who underwent TTE and CMR imaging. Ventricular and atrial diameter measurements from TTE-derived standard parasternal long axis and apical four-chamber views and CMR-derived three- and four-chamber views were recorded. The concordance between the two methods was compared using intra-class correlation coefficients (ICC) and Bland–Altman plots. *Results:* Of the patients, 25 (61%) were male and the mean age was 48.12 ± 16.79. The mean ICC for LVDD between CMR observers was 0.957 (95% CI: 0.918–0.978), while the mean ICC between CMR and TTE measurements were 0.849 (95% CI: 0.709–0.922) and 0.836 (95% CI: 0.684–0.915), respectively. The mean ICC for the right ventricle between CMR observers was 0.985 (95% CI: 0.971–0.992), while the mean ICC between CMR and TTE measurements were 0.869 (95% CI: 0.755–0.930) and 0.892 (95% CI: 0.799–0.942), respectively. Passing–Bablok Regression and Bland–Altman plots indicated high concordance between the two methods. *Conclusions:* TTE and CMR indicated high concordance in chamber diameter measurements for which the CMR should be considered in patients for whom optimal evaluation with TTE could not be performed due to their limitations.

## 1. Introduction

Imaging methods are the most important tools that guide the clinician in the diagnosis and treatment of heart disease [1]. Transthoracic echocardiography (TTE) is the first-choice imaging technique in the daily practice of cardiology as it is non-invasive, low-cost, easily available, reproducible, radiation-free, requires short examination time, allowes per-patient application and provides real-time imaging in different plans [2]. The disadvantages of TTE such as requiring operator dependency and experience, difficulties in performing optimal evaluation due to body structure, obesity or chronic lung diseases or limited spatial resolution with a narrow field of view increases the need for different imaging modalities, which allow the evaluation of the structure and function of the heart day by day [3,4].

CMR is a non-invasive imaging method which is widely used in the evaluation of cardiac functions because of its excellent temporal and spatial resolution, lack of ionizing radiation, less operator dependency and lack of limitations such as acoustic window problem [5,6,7]. CMR is particularly considered to be the gold standard method for the volumetric assessment of ventricular functions, mass measurement, and detection of myocardial scarring [1,3,4]. Moreover, in recent years, it has been reported that CMR has received new indications in the assessment of myocardial edema, iron overload and fibrosis, and the diagnosis of congenital heart disease and heart failure [4,5]. Studies investigating whether the data measurements obtained from TTE and CMR, which are the two different imaging modalities commonly used in the diagnosis of heart disease, are in concordance with each other generally focus on the volumetric measurements (left ventricular end-diastolic volume, end-systolic volume, stroke volume, cardiac output, ejection fraction) and left ventricular muscle mass [6,7,8]. However, many heart diseases with primary or secondary causes are known to affect the heart chamber size directly or indirectly [9,10]. There is a limited number of studies in the literature regarding whether TTE and CMR provide data in concordance with each other in terms of the heart chamber measurements, and these studies have only investigated the compliance of the left ventricular systolic and diastolic diameter measurements [9,10,11].

The aim of this study is to investigate the imaging data of patients who underwent TTE and CMR with different indications due to known or suspected heart disease and to determine whether the diameter measurement values of all cardiac chambers are in concordance with each other according to both imaging methods.

## 2. Materials and Methods

### 2.1. Subject Selection and Imaging Protocols

The study was carried out on the basis of retrospective examination of the patients who underwent TTE and CMR between 1 January 2018 and 30 September 2018. A total of 41 patients, four patients with investigation of fibrosis in heart failure, nine patients with evaluation of congenital heart disease and 28 patients evaluated with suspected myocarditis, were included in the study. The echocardiographic data of the cardiac chamber measurements of all patients were obtained from echocardiography reports which were performed by an experienced cardiologist (C) and registered to the hospital information system. Measurements of all patients were performed by the same cardiologist. An experienced radiologist (R1, seven years’ experience of CMR studies) examined the whole study group. A second radiologist (R2, three years’ experience of CMR) also examined the same study group. CMR data of cardiac chamber measurements of patients were obtained from the independent examination results performed manually via the Picture Archiving and Communication System (PACS). The mean time between the application of the two imaging modalities was 13.8 ± 6.2 days and the patients with more than 1 month elapse between these two imaging methods were not included in the study. The TTE evaluation in all patients was operated with Vivid 7, GE Vingmed Ultrasound, Horten, Norway (application software version: 6.1.3) using a 2.5–3.5 MHz probe in the left lateral position, parasternal long axes and apical four-chamber images were obtained. To improve image quality, depth, focus position, frame rate and sector size was adjusted. The measurements of the left ventricle were carefully measured at the parasternal long-axis window, perpendicular to the long axis of the left ventricle and at the level of the mitral valve leaflet tips. The end-diastolic and end-systolic frames were identified based on the maximal and minimal left ventricle cavity size. Left atrial anteroposterior diameter measurements were made from the parasternal long axis window, perpendicular to the distance between the aortic sinus level and the left atrium posterior wall. The chamber measurements of the right ventricle and right atrium were performed from the apical four-chamber view window. The basal diameter of the right ventricle was measured at one third of the basal of the right ventricle. The diameter of the atrium was measured at the distance between the inter-atrial septum and the lateral wall. The diameter of cardiac cavities was measured based on the criteria suggested by the American Society of Cardiovascular Imaging and the European Echocardiography Association [12] (Figure 1A–D). CMR measurements were performed as follows; while patients were examined with a 1.5 Tesla MRI unit (Magnetom Aera, Siemens, Erlangen, Germany) and retrospective electrocardiographic triggering, eight-channel cardiac coils were used for image acquisition. The position and orientation of the heart in the thorax were determined after the serial thoracic reference images were taken. With the steady-state free precession (SSFP) sequences, the left ventricular two-chamber image was obtained by holding the breath for 10–15 s at the end of expiration (imaging parameters: repetition time 3.6 ms; echo time 1.6 ms; 350 mm imaging area; 6 mm section thickness; with 2 mm gap; 45° flip angle; 14 views per segment (VPS); matrix 224 160). Then, four-chamber short axes and three-chamber SSFP sequences were obtained. The images were evaluated on the workstation (Sectra IDS7, Linkoping, Sweden) and measurements of both ventricles and atrium were performed [13] (Figure 2A–D). The diameter of the right ventricular end-diastolic (RVED) was measured between the right ventricular endocardium and the interventricular septum, parallel to the tricuspid valve and 1 cm distal from the valve on four-chamber steady-state free precession (SSFP) sine images [14]. Whereas, the diameter of the left ventricular end-diastolic (LVED) and left ventricular end-systolic (LVES) were measured between the left ventricular endocardium and the interventricular septum, parallel to the mitral valve and 1 cm distal from the valve on three-chamber SSFP sine images. The mediolateral diameter of the right atrium was measured in the mid-section of the atrium between the right atrial free wall and the interatrial septum, parallel to the tricuspid valve. The right atrial diameter measurement was calculated during the ventricular end-systolic phase on four-chamber SSFP sine images. On the other hand, the left atrial anteroposterior diameter was measured during ventricular end-systolic phase on three-chamber SSFP sine images from left atrial free wall to the interatrial septum, parallel to the mitral valve. Care was taken to measure atrial and ventricular diameters at the same anatomic landmarks in CMR and TTE [15]. Both of the radiologists were blinded to each other’s result as well as the TTE results. Approval of Trakya University Faculty of Medicine Scientific Research Ethics Committee was received to conduct the study (TÜTF-BAEK 2018/382).

### 2.2. Statistical Examinations

The normal distribution hypothesis was tested with the Shapiro–Wilk test. In the evaluation of data, frequency, percentage and mean were used as descriptive statistical methods. In order to evaluate the correlation between TTE and CMR measurements, Pearson’s correlation coefficient and intra-class correlation (ICC) coefficient were calculated. Passing–Bablok regression analysis was performed to evaluate the concordance between TTE and CMR measurements and Bland–Altman plots with 95% limits of agreement (mean difference ± 1.96 standard deviation of the difference) were created for pairs of measurements to evaluate any deviations between observers [16]. The significance level was accepted as 0.05 in all statistical analyses. All statistical analyses were conducted using TURCOSA (Turcosa Analytics Ltd. Co, Kayseri/Turkey, www.turcosa.com.tr) statistical software.

## 3. Results

Of the 41 patients included in the study, 25 were male (61%) and the mean age of the patients was 48.12 ± 16.79. Table 1 shows the mean values of the chamber measurements of all three observers. A Bonferroni test was used as a multiple comparison test after one way variance analysis. For left ventricular end-diastolic (LVED) diameter and left ventricular end-systolic (LVES) diameter measurements, there was a significant difference between R1 and R2 (respectively, *p* < 0.001 and *p* = 0.046). There was a significant difference between R1–C for left atrial anteroposterior (LAAP) diameter measurement (*p* = 0.04). There was a significant difference between R1–C and R2–C for right ventricle (RV) and right atrium (RA) measurements (*p* < 0.001). Among LVED, LVES and LAAP diameter, RV and RA diameters measured by R1–R2, R1–C, R2–C, when the intraclass correlation coefficients were examined according to the diameter measurements of cardiac chambers, it was seen that there was a good level of agreement between all three observers (Table 2). When Pearson correlation coefficients between the three observers were examined, it is seen that there was a good concordance in a positive direction between diameter measurements by R1–R2, R1–C and R2–C (Table 3). Scatter plots between observers for diameter measurements are given in Figure 3. When the results of Passing–Bablok regression analysis were examined for each observer, LVED, LVES, left atrium (LA), RV and RA diameter measurements were found to be in concordance. In all regression analysis results, the cut-off point had the value of 0, while the slope had the value of 1. This observation shows that there was a good level of concordance between the observers (Appendix A). Moreover, when Bland–Altman plots were analyzed (Appendix A), it was seen that there was a random distribution around the mean difference in measurements. This shows that there was a concordance between the observers.

## 4. Discussion

The most important findings obtained from this study can be listed as follows: (1) The data for the LVED, LVES and LAAP diameter measurements from the parasternal long axis view obtained with TTE were in concordance with the data from the three-chamber view obtained with CMR, (2) the data for the RV and RA diameter measurement values from the apical four-chamber view obtained with TTE were numerically lower than the data from the four-chamber view obtained with CMR, while the concordance still exists.

In clinical practice, the most frequently evaluated parameters with TTE are ventricular systolic and diastolic functions, valve structures and functions, ventricular mass and diameter of the heart chambers [12]. Accurate and reproducible diameter measurements of the chambers are important indicators in the diagnosis of the underlying disease, determining its severity, evaluating the effectiveness of the treatment and defining the prognosis [2,9]. In a study evaluating the data obtained from TTE and CMR diameter measurements, it was reported that the concordance between LVED and LVES diameters was high in cardiomyopathy patients [9]. Similarly, in our study, regarding the LVED and LVES diameter measurements, it was found that the concordance between the two modalities was high. Again, in terms of LVED and LVES diameters, there was a high level of concordance between the data obtained from the three-chamber view with CMR and the data obtained from the parasternal long axis window with TTE. In the literature, the high concordance between LVED and LVES diameter measurements with TTE and CMR is explained by the similarity of the imaging views used in the two methods [9,10,11,12]. The diameter and volume of LA are considered as important prognostic indicators in the general population and in the presence of heart failure, acute myocardial infarction, cardiomyopathy and mitral regurgitation [17]. Furthermore, LA functions are a reflection of left ventricular diastolic performance [17]. When the data of LAAP diameter measurements were examined, it was determined that the data obtained from parasternal long axis, which is the standard measurement view for LAAP in TTE, was highly in concordance with the data obtained from the three-chamber view in CMR. The similarity of the imaging windows used in the two methods can be considered as the most important reason for the concordance observed in these measurement values. The structural and functional characteristics of the right ventricle are important factors in prognosis in various heart diseases, especially in congenital heart diseases, pulmonary hypertension, myocardial infarction and arrhythmogenic right ventricular cardiomyopathy [18,19,20]. The most commonly used echocardiographic view for the evaluation of the size and function of the right ventricle is apical four-chamber view [21]. However, it is not always possible to evaluate the right ventricle with echocardiography because of its asymmetric and highly variable shape, thinner myocardial wall thickness, retrosternal localization or due to difficulties in obtaining an optimal view from patient to patient [10,22]. In addition to these limitations, using the measurement values obtained from the apical four-chamber view for the assessment of the size and functions of the right ventricle and right atrium in daily practice, results in the detection of lower values [2,23]. In our study, this may be the most probable cause of the numerical difference between right atrial and right ventricular measurements obtained with echocardiography and apical four-chamber view obtained with CMR. In the consensus report of the American Society of Echocardiography and the European Society of Cardiovascular Imaging, the use of focused right ventricular image (RV-focused view) rather than the apical four-chamber image is recommended for the measurement of right ventricular size and function [2]. CMR allows for more accurate results, as many limitations in echocardiography are ruled out in the evaluation of right ventricular size and function, allowing the measurement to be made from a study area similar to the focused right ventricular imaging view. The concordance between TTE and CMR values for the measurement of right heart chambers of the patients included in our study can be explained by the fact that the numerically determined difference was not large enough to affect the clinical decision. CMR allows for the calculation of right ventricular systolic function, volumes and myocardial mass, especially by drawing the endocardial and epicardial borders from the short axis images of the heart and correlating these drawings with long axis images (four chambers and two chambers) [18,19,20]. For this reason, CMR is accepted as the gold standard in the evaluation of the right ventricle [10,18,19,20]. Therefore, the CMR option should be kept in mind when the values for size and function of the right ventricle limit clinical decision-making or when the optimal evaluation cannot be made in echocardiography.

### Limitations

The findings of this study should be interpreted with some limitations. In addition to being a retrospective and cross-sectional study, due to the lack of data evaluation according to age and gender, the findings obtained from the study will not reflect the general population. The use of data from a single center and relatively few patients is another limitation. In addition, numerical differences between TTE and CMR, especially in the right heart chambers, may be remarkable. The absence of comparison for these differences can be considered as a limitation. However, it should not be overlooked that the aim of the study was to determine whether there is a general concordance rather than whether there are any numerical differences between the two methods. Lastly, another limitation of the study was that data from a single cardiologist of TTE measurements were used, although data from two different radiologists from CMR measurements were used.

## 5. Conclusions

As a result, this study revealed that left heart chambers show a high level of concordance between the parasternal long axis in TTE and the three-chamber view in CMR. Of note, concordance between TTE and CMR as demonstrated with statistical analysis also persists in the evaluation of RV chambers. These findings substantiate the clinical efficacy of CMR in clinical decision-making for the diagnosis, treatment as well as follow-up of certain cardiac diseases, and hence; might potentially suggest this modality as an important substitute for TTE particularly in the setting of certain challenges including poor echocardiographic image quality.

## Figures and Tables

**Figure 1 medicina-55-00260-f001:**
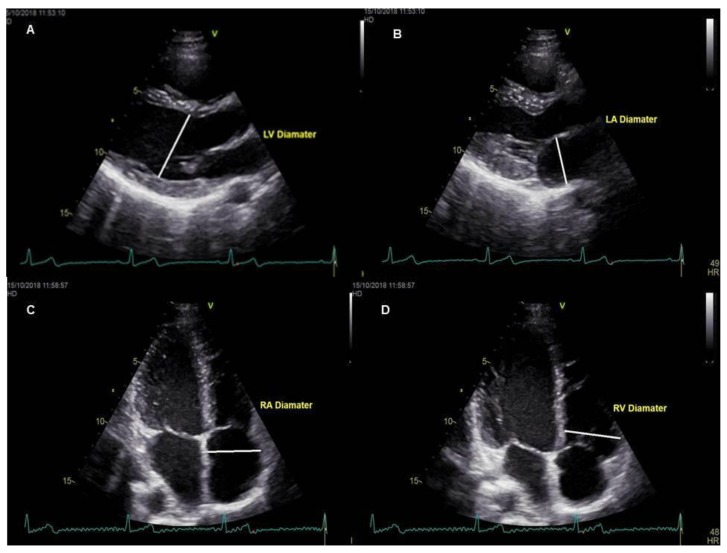
Measurements obtained from echocardiography views. (**A**): Parasternal long-axis left ventricular diastolic diameter, (**B**): parasternal long-axis left atrium diameter, (**C**): right atrial diameter from apical four-chamber view, (**D**): right ventricular diameter from apical four-chamber view.

**Figure 2 medicina-55-00260-f002:**
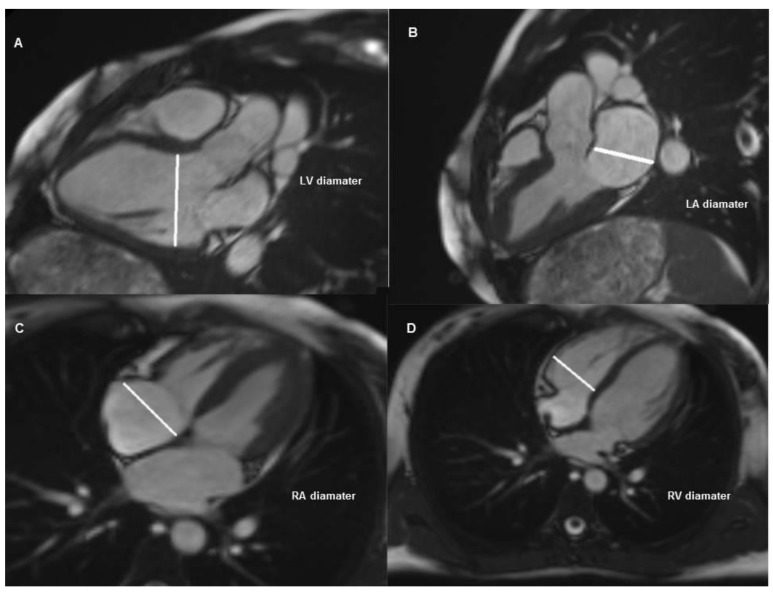
Measurements from cardiac magnetic resonance imaging. (**A**): Left ventricular diastolic diameter from three-chamber view, (**B**): left atrial diameter from three-chamber view, (**C**): right atrial diameter from four-chamber view, (**D**): right ventricular diameter from four-chamber view.

**Figure 3 medicina-55-00260-f003:**
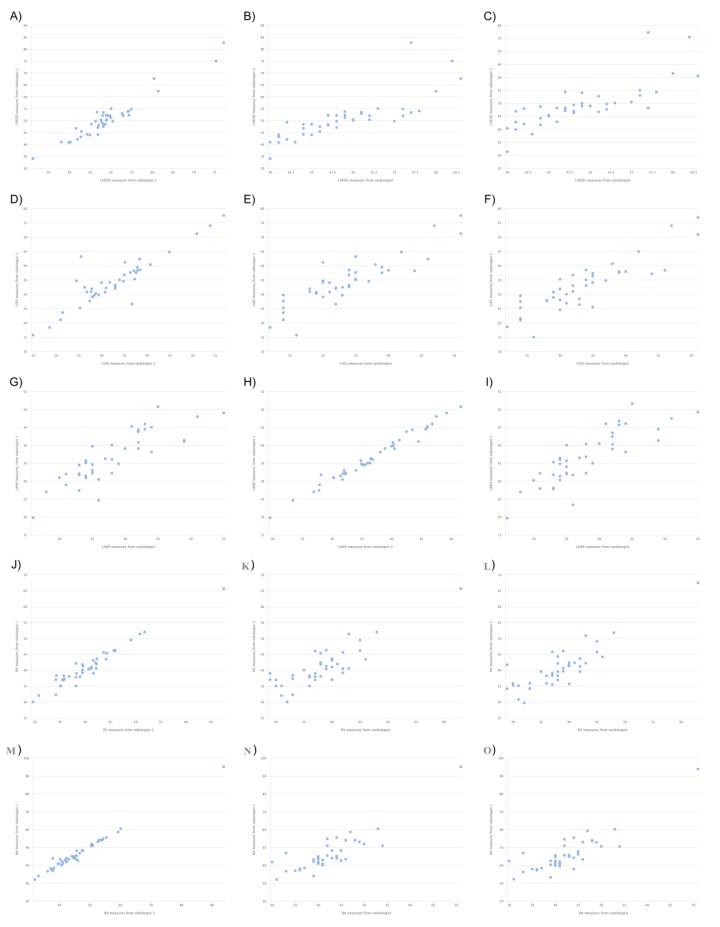
Scatter plots between CMR and TTE measurements. (**A**) LVED measures between R1 and R2. (**B**) LVED measures between R1 and C. (**C**) LVED measures between R2 and C. (**D**) LVES measures between R1 and R2. (**E**) LVES measures between R1 and C. (**F**) LVES measures between R2 and C. (**G**) LAAP measures between R1 and R2. (**H**) LAAP measures between R1 and C. (**I**) LAAP measures between R2 and C. (**J**) RV measures between R1 and R2. (**K**) RV measures between R1 and C. (**L**) RV measures between R2 and C. (**M**) RA measures between R1 and R2. (**N**) RA measures between R1 and C. (O) RA measures between R2 and C.

**Table 1 medicina-55-00260-t001:** Mean values of chamber diameter measurements.

Chambers	R1	R2	C	*p* Value		
	Mean ± SD	Mean ± SD	Mean ± SD	R1–R2	R1–C	R2–C
LVED	51.81 ± 8.72	49.24 ± 8.33	48.92 ± 6.13	<0.001 *	0.075	1.000
LVES	34.43 ± 8.47	32.89 ± 8.64	33.56 ± 7.23	0.046 *	0.531	0.922
LAAP	36.48 ± 6.93	36.60 ± 7.29	37.97 ± 6.30	1.000	0.040 *	0.089
RV	41.03 ± 6.27	40.57 ± 6.55	38.26 ± 6.19	0.192	<0.001 *	<0.001 *
RA	46.58 ± 10.25	46.15 ± 10.19	42.49 ± 7.08	0.181	<0.001 *	<0.001 *

* Statistical significance (*p* < 0.05); abbreviations: R1: Radiologist 1, R2: Radiologist 2, C: Cardiologist, LVED: left ventricular end-diastolic diameter, LVES: left ventricular end-systolic diameter, LAAP: left atrial anteroposterior, RV: right ventricle, RA: right atrium.

**Table 2 medicina-55-00260-t002:** Concordance of chamber diameter measurements between observers.

Chambers	R1–R2		R1–C		R2–C	
	ICC	95% CI	ICC	95% CI	ICC	95% CI
LVED	0.957	0.918–0.978	0.849	0.709–0.922	0.836	0.684–0.915
LVES	0.933	0.870–0.965	0.929	0.863–0.963	0.931	0.867–0.964
LAAP	0.994	0.989–0.997	0.903	0.816–0.947	0.901	0.788–0.943
RV	0.985	0.971–0.992	0.869	0.755–0.930	0.892	0.799–0.942
RA	0.995	0.99–0.007	0.838	0.687–0.916	0.841	0.694–0.918

Abbreviations: ICC: intra-class correlation coefficient, CI: confidence interval, R1: Radiologist 1, R2: Radiologist 2, C: Cardiologist, LVED: left ventricular end-diastolic diameter, LVES: left ventricular end-systolic diameter, LAAP: left atrium anteroposterior, RV: right ventricle, RA: right atrium.

**Table 3 medicina-55-00260-t003:** Correlation coefficients between the observers.

Chambers	R1–R2		R1–C		R2–C	
	*r*	*p* *	*r*	*p* *	*r*	*p* *
LVED	0.942	<0.0001	0.808	<0.0001	0.752	<0.0001
LVES	0.890	<0.001	0.881	<0.001	0.886	<0.001
LAAP	0.989	<0.001	0.847	<0.001	0.844	<0.001
RV	0.972	<0.001	0.851	<0.001	0.863	<0.001
RA	0.991	<0.0001	0.870	<0.0001	0.854	<0.0001

* Statistical significance (*p* < 0.05); abbreviations: R1: Radiologist 1, R2: Radiologist 2, C: Cardiologist, LVED: left ventricular end-diastolic diameter, LVES: left ventricular end-systolic diameter, LAAP: left atrium anteroposterior, RV: right ventricle, RA: right atrium.

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
