# Peer review of "Cardiac Magnetic Resonance Imaging and Transthoracic Echocardiography: Investigation of Concordance between the Two Methods for Measurement of the Cardiac Chamber"

_medicina, 2019, doi:10.3390/medicina55060260_

Round 1

Reviewer 1 Report

This manuscript presented a comparison of measurements of heart chambers diameters using two imaging techniques: TTE and cardiac MRI. The study cohort included 41 human subjects who underwent echocardiographic imaging and MRI on separate days. The MRI data were analyzed by two radiologists while the echo data was analyzed by one sonographer. A series of statistical analyses were performed to examine the concordance of diameters of the left ventricle, left atrium, right ventricle, and right atrium. Investigating the difference in echo and MRI measurements is a long-standing topic and there exist a large number of publications.  Conventional comparison has been focusing on volumetric measurements such as ejection fraction and cavity volumes. This study focused on the diameters of the four chambers which has not been extensively covered by the literature. There are a number of comments for the authors to address. 

1. Introduction

This section needs to be re-organized. In fact that the second paragraph of the Discussion section should be incorporated into the introduction as it is trying to justify the necessity of this study. 

"concordance" should be replaced with "in concordance" throughout the entire manuscript. 

2.1 Subject Selection and Imaging Protocols

A summary of the patients' known or suspected diseases should be included in this section.  

The authors need to provide details on how the diameter measurements were made by the observers, whether it's using the machine built-in analysis tools or third party software. Detailed need to be included. 

When making diameter measurement, is it very important to ensure the location at which the measurement was the same between the two sets of imaging data. How was this determined in this study? 

For the CMR data, were two radiologists analyzing the same data or two sets of CMR data? If yes, why was this set-up not available for the echo case. There should be a possibility for a second sonographer to analyze the data performed by another sonographer? 

3. Results:

The Bland-Altman graphs showed some very significant outliers. Did the authors investigate these outliers? 

All of the results presented in this study rely on the fact that the measurements from echo and CMR were made as consistent as possible. As the reviewer pointed out above, if the locations at which the measurements made differed between the two imaging modalities, then this places significant doubt on the results. The authors should careful justify this in the revised manuscript. 

As many studies have looked at the difference in volumetric measurements. Have the authors attempted to look at these metrics in this study too? Adding the volumetric measurements will significantly strengthen this study. 

Author Response

Response to Reviewer 1 Comments

We truly appreciate all the comments that the reviewer has made, and based on your recommendations the pertinent changes have been made.  Bellow you can find the responses addressing the specific concerns of the reviewers.

1. Introduction

Point 1: This section needs to be re-organized. In fact that the second paragraph of the Discussion section should be incorporated into the introduction as it is trying to justify the necessity of this study. 

Response 1:  Second paragraph of the discussion section was combined with introduction and introduction section was re-organized.

Point 2: "concordance" should be replaced with "in concordance" throughout the entire manuscript. 

Response 2:  Throughout the entire manuscript "concordance" was replaced with "in concordance".

2.1 Subject Selection and Imaging Protocols

Point 3:  A summary of the patients' known or suspected diseases should be included in this section.  

Response 3: A summary of the patients were given in this section.  

Point 4: The authors need to provide details on how the diameter measurements were made by the observers, whether it's using the machine built-in analysis tools or third party software. Detailed need to be included. 

Response 4:  CMR data analyzes were evaluated by radiologists in Sectra, the Picture Archiving and Communication System (PACS) system, and measurements were performed manually. Data analysis on echocardiography was performed manually by the cardiologist using Vivid 7 Pro, GE Vingmed Ultrasound, Horten, Norway (Application software version: 6.1.3).

A detailed description is added to the "2.1. Subject Selection and Imaging Protocols" section of the article.

Point 5: When making diameter measurement, is it very important to ensure the location at which the measurement was the same between the two sets of imaging data. How was this determined in this study? 

Response 5: Descriptive information on how to measure diameters in accordance with the recommendations has been added to the "2.1. Subject Selection and Imaging Protocols" section of the article.

Point 6: For the CMR data, were two radiologists analyzing the same data or two sets of CMR data? If yes, why was this set-up not available for the echo case. There should be a possibility for a second sonographer to analyze the data performed by another sonographer? 

Response 6:  The section on obtaining TTE and CMR data was revised in detail and added to the article. Due to the retrospective nature of the study, echocardiographic data of the patients were obtained from the hospital report recording system. Therefore, it was not possible to perform measurements by a second sonograph. However, CMR images of the patients could be evaluated by two radiologists because they were registered in the hospital PACS system.  This limitation was added to the relevant section of the study.

3. Results:

Point 7: The Bland-Altman graphs showed some very significant outliers. Did the authors investigate these outliers? 

Response 7: We have investigated the outliers in the Bland-Altman plot and inspected these subjects’ measurements. We have concluded that there are no any errors in the measurements, therefore these outlying measures are caused by nature of the measurements.

Point 8: All of the results presented in this study rely on the fact that the measurements from echo and CMR were made as consistent as possible. As the reviewer pointed out above, if the locations at which the measurements made differed between the two imaging modalities, then this places significant doubt on the results. The authors should careful justify this in the revised manuscript. 

Response 8: We thank you for the input. As we added in materials and methods section in revised manuscript care was taken to measure atrial and ventricular diameters at the same anatomic landmarks in CMR and TTE.

Point 9: As many studies have looked at the difference in volumetric measurements. Have the authors attempted to look at these metrics in this study too? Adding the volumetric measurements will significantly strengthen this study. 

Response 9: In this study, we aimed to investigate the in concordance of two different imaging modalities (TTE and CMR) in cardiac chamber measurements. Due to the missing volumetric measurements in some of the patients in the TTE data, we could not include the in concordance between volumetric measurements in our study. However, your suggestions will be guiding us for future prospective studies.

Reviewer 2 Report

Medicina-483662 ‘Cardiac magnetic resonance imaging and transthoracic echocardiography: investigation of concordance between the two methods for measurement of cardiac chambers.’ First author: Muhammet Gürdoğan.

I would like to thank all authors for their work and contribution. In a current study, investigators compared two different imaging modalities – echocardiography and cardiac MRI and their accuracy and concordance to estimate ventricular and atrial diameters. Manuscript is short, easy to read; however, the importance of the results is limited as clinical decisions in daily routine are usually based on volumetric or functional parameters, and only rarely on diameters.

Please, have my comments and questions bellow:

1. Abstract is missing and needs to be added.

2. Keywords and abbreviations are missing and must be added as well.

3. Please, always explain abbreviation in the text when you use it for the first time (TTE: page 2, line 31; CMR: page 2, line 36).

4. The description of study population is missing. Please, explain how participants were selected. Were there any inclusion / exclusion criteria?

5. Why echocardiographic images were analyzed one time (by cardiologist) and CMR images two times (by radiologist)? Can authors explain it?

6. What software was used to analyze echo data?

7. Table 1 demonstrates mean values ± SD of ventricular and atrial diameters. Please, add P values for compared pairs: R1 vs. R2, R1 vs. C, R2 vs. C.

8. Results must be rewritten including numbers and P values.

9. Statement ‘<…> it was found that the concordance between two radiologists was high <…>’ (page 4, line 131-132) is misleading. This is comparison of two modalities and not of two different specialists (radiologist vs. cardiologist). Please, adjust accordingly.

10. Please, rewrite conclusions. It must be adjusted according to study findings, but not something general.

11. Please, prepare nice correlation Figure of two techniques and make Table 4 as supplemental Table.

Author Response

Response to Reviewer 2 Comments

We truly appreciate all the comments that the reviewer has made, and based on your recommendations the pertinent changes have been made.  Bellow you can find the responses addressing the specific concerns of the reviewers.

Point 1. Abstract is missing and needs to be added.

Response 1: Abstract section added.

Point 2. Keywords and abbreviations are missing and must be added as well.

Response 2: Keywords added to the relevant section.  The abbreviations are included in the article and below the table.

Point 3. Please, always explain abbreviation in the text when you use it for the first time (TTE: page 2, line 31; CMR: page 2, line 36).

Response 3: Abbreviations are explained in accordance with the suggestion.

Point 4. The description of study population is missing. Please, explain how participants were selected. Were there any inclusion / exclusion criteria?

Response 4:  The clinical characteristics of the patients included in the study in accordance with your recommendations are explained in section  "Subject Selection and Imaging Protocols"

Point 5. Why echocardiographic images were analyzed one time (by cardiologist) and CMR images two times (by radiologist)? Can authors explain it?

Response 5:  The section on obtaining TTE and CMR data was revised in detail and added to the article. Due to the retrospective nature of the study, echocardiographic data of the patients were obtained from the hospital report recording system. Therefore, it was not possible to perform measurements by a second sonograph. However, CMR images of the patients could be evaluated by two radiologists because they were registered in the hospital PACS system.  This limitation was added to the relevant section of the article.

Point 6. What software was used to analyze echo data?

Response 6: Data analysis in echocardiography was performed using ‘’Vivid 7 Pro, GE Vingmed Ultrasound, Horten, Norway (Application software version: 6.1.3).’’  This information  was added to the relevant section in the article.

Point 7. Table 1 demonstrates mean values ± SD of ventricular and atrial diameters. Please, add P values for compared pairs: R1 vs. R2, R1 vs. C, R2 vs. C.

Response 7: P values were added to Table 1 in accordance with the suggestion.

Point 8. Results must be rewritten including numbers and P values.

Response 8: The results of Table 1 were rewritten in the relevant section.

Point 9. Statement ‘<…> it was found that the concordance between two radiologists was high <…>’ (page 4, line 131-132) is misleading. This is comparison of two modalities and not of two different specialists (radiologist vs. cardiologist). Please, adjust accordingly.

Response 9: We revised this sentence.

Point 10. Please, rewrite conclusions. It must be adjusted according to study findings, but not something general.

Response 10: Conclusions section were revised.

Point 11. Please, prepare nice correlation Figure of two techniques and make Table 4 as supplemental Table.

Response 11: We have prepared scatter plots for measurements between measurers. The plot is added as Figure 3. Table 4 is given as supplementary table (Table S1).

Round 2

Reviewer 2 Report

Authors adjusted manuscript according to the previous comments and suggestions and improved their manuscript.

Please, have my comments bellow:

1. page 2, line 28: ‘<…> volumetric ventricular functions <…>’. This phrase needs to be rewritten as ‘ventricular volumes’ or ‘ventricular function’.

2. Please, cite the important manuscript published in Medicina as a first reference in the manuscript:

     ‘Lapinskas T. Ischemic heart disease: a comprehensive evaluation using cardiovascular magnetic resonance. Medicina (Kaunas) 2013,49,97-110’.

3. Please, explain abbreviations when use it in the text for the first time (RVED: page 5, line 135; LVED and LVES: page 5, line 138).

4. Please, add the agreement range of Bland-Altman analysis in the ‘Statistical examinations’ section with the appropriate citation.

5. Please, shortly mention in the text experience of the investigators as the agreement in LVED and LVES measurements derived from CMR by two different radiologists is significant.

6. Please, be clear in the ‘Discussion’ that despite significant difference in absolute values between two techniques, agreement is excellent. This difference is due to intrinsic differences in imaging modalities. CMR provides better quality images, the delineation of endocardial surface is much better. The TTE images used for the estimation of right heart diameters were not RV-focused and might be a reason for significant differences and slightly lower agreement between the techniques. The main message must focus about the AGREEMENT of measurements, but not ABSOLUTE VALUES! Please, adjust second sentence in the ‘Conclusions’ accordingly.

7. ‘<…> In the future, studies performed with large-scale patient populations with the use of devices from multi-center and different companies may result in demonstrating the importance of CMR in volumetric evaluation as well as in diameter measurement <…>’ (page 8, line 260-263). Please, remove this sentence from ‘Conclusions’.

8. Please, add legend of figure 3.

9. The legend of Table 1. Please, change ‘Radiolog’ to ‘Radiologist’.

10. English proofing is highly recommended!

Author Response

Response to Reviewer 2 Comments

Many thanks for your didactic and comprehensive evaluation of the manuscript. Below, you will find our responses:

Point 1. page 2, line 28: ‘<…> volumetric ventricular functions <…>’. This phrase needs to be rewritten as ‘ventricular volumes’ or ‘ventricular function’.

Response 1: As per your suggestion, the relevant sentence has been revised.

Point 2. Please, cite the important manuscript published in Medicina as a first reference in the manuscript:

‘Lapinskas T. Ischemic heart disease: a comprehensive evaluation using cardiovascular magnetic resonance. Medicina (Kaunas) 2013,49,97-110’.

Response 2: As per your suggestions, new reference has been added to the İntroduction section.

References section has been revised.

Point 3. Please, explain abbreviations when use it in the text for the first time (RVED: page 5, line 135; LVED and LVES: page 5, line 138).

Response 3: As per your suggestions, abbreviations has been  explained.

Point 4. Please, add the agreement range of Bland-Altman analysis in the ‘Statistical examinations’ section with the appropriate citation.

Response 4: The following sentence has been added to  Statistical Examinations section.

"Bland-Altman plots with 95% limits of agreement (mean difference ± 1.96 standard deviation of the difference) were created for pairs of measurements to evaluate any deviations between observers."

Point 5. Please, shortly mention in the text experience of the investigators as the agreement in LVED and LVES measurements derived from CMR by two different radiologists is significant.

Response 5: As per your suggestion, experience of the investigators has been added in the Materials and Methods section.

"An experienced radiologist (R1, 7 years’ experience of CMR studies) examined the whole study group. A second radiologist (R2, 3 years’ experience of CMR) also examined same study group."

Point 6. Please, be clear in the ‘Discussion’ that despite significant difference in absolute values between two techniques, agreement is excellent. This difference is due to intrinsic differences in imaging modalities. CMR provides better quality images, the delineation of endocardial surface is much better. The TTE images used for the estimation of right heart diameters were not RV-focused and might be a reason for significant differences and slightly lower agreement between the techniques. The main message must focus about the AGREEMENT of measurements, but not ABSOLUTE VALUES! Please, adjust second sentence in the ‘Conclusions’ accordingly."

Response 6: As per your suggestions, the Conclusion section has been revised.

"Of note, concordance between TTE and CMR as demonstrated with statistical analysis also persists in the evaluation of RV chambers. These findings substantiate the clinical efficacy of CMR in clinical decision-making for the diagnosis, treatment as well as follow-up of certain cardiac diseases, and hence; might potentially suggest this modality as an important substitute for TTE particularly in the setting of certain challenges including poor echocardiographic image quality."

Point 7. ‘<…> In the future, studies performed with large-scale patient populations with the use of devices from multi-center and different companies may result in demonstrating the importance of CMR in volumetric evaluation as well as in diameter measurement <…>’ (page 8, line 260-263). Please, remove this sentence from ‘Conclusions’.

Response 7:  As per your suggestion, relevent sentence has been removed from the  Conclusions section.

Point 8. Please, add legend of figure 3.

Response 8: As per your suggestion, legend has been added to the Figure 3 as follow:

"Figure 3: Scatter plots between CMR and TTE measurements. A) LVEDD measures between R1 and R2. B) LVEDD measures between R1 and C. C) LVEDD measures between R2 and C. D) LVES measures between R1 and R2. E) LVES measures between R1 and C. F) LVES measures between R2 and C. G) LAAP measures between R1 and R2. H) LAAP measures between R1 and C. I) LAAP measures between R2 and C. J) RV measures between R1 and R2. K) RV measures between R1 and C. L) RV measures between R2 and C. M) RA measures between R1 and R2. N) RA measures between R1 and C. O) RA measures between R2 and C."

Point 9. The legend of Table 1. Please, change ‘Radiolog’ to ‘Radiologist’.

Response 9: As per your suggestion, the relevant word has been changed.

Point 10. English proofing is highly recommended!

Response 10: As per your suggestion, the whole article has been revised in terms of typographical errors, grammar and meaning shifts in English. (Particularly in the introduction and material method sections, parts indicated with blue color are re-written.)
